# FaceVAE: Generation of a 3D Geometric Object Using Variational Autoencoders

**Sungsoo Park [1,2]** and **Hyeoncheol Kim [2,*]**

1   Korea Virtual Reality Inc., Seoul 05719, Korea; littlepu@naver.com
2   Department of Computer Science and Engineering, Korea University, Seoul 02841, Korea
*   Correspondence: hkim64@gmail.com

**Abstract:** Deep learning for 3D data has become a popular research theme in many fields. However, most of the research on 3D data is based on voxels, 2D images, and point clouds. At actual industrial sites, face-based geometry data are being used, but their direct application to industrial sites remains limited due to a lack of existing research. In this study, to overcome these limitations, we present a face-based variational autoencoder (FVAE) model that generates 3D geometry data using a variational autoencoder (VAE) model directly from face-based geometric data. Our model improves the existing node and edge-based adjacency matrix and optimizes it for geometric learning by using a face- and edge-based adjacency matrix according to the 3D geometry structure. In the experiment, we achieved the result of generating adjacency matrix information with 72% precision and 69% recall through end-to-end learning of Face-Based 3D Geometry. In addition, we presented various structurization methods for 3D unstructured geometry and compared their performance, and proved the method to effectively perform reconstruction of the learned structured data through experiments.

**Keywords:** VAE; deep learning; 3D geometry; graph data; generation model

## 1. Introduction

Deep learning for 3D data is being researched in various fields because of its great utility. Moreover, 3D data exist in various forms, such as 2D images, voxels, point clouds, and polygon models, and deep learning research is being conducted based on these various data. However, as the research is being conducted mainly on 2D images [1–6], voxels [1,7–11], and point clouds [12–14], which are relatively easy to learn, there is a limit to directly applying the research results in actual industrial sites as the commonly used 3D data format is polygon based data [15]. In order to overcome this limitation, research on direct polygon-based data is required [16–18]. In this study, in order to overcome this limitation, research on a generation model of 3D data was conducted directly through polygon data.

In deep learning about graphs, much research has been conducted in the fields of social networks [19–22], chemistry [23,24], medicine [25], and computer vision [6,18,25], among others, and many achievements have been made. Deep learning's effectiveness has been proven in link prediction and label discrimination [19–22]. Since the face data constituting a polygon basically take the form of graph data, they can be considered closely related to the study of graph data.

In this research, based on the adjacency matrix and feature matrix, which are graph data structures, we propose an adjacent matrix face- and edge-optimized for 3D geometric data by improving the structure of graph data. In addition, we achieved the result of generating adjacency matrix information with 72% precision and 69% recall through end-to-end learning with vertex position and face index data, which are basic geometries.

## 2. Related Work

### 2.1. 3D Data

Deep learning for 3D data is being studied in many fields, and because of its dimensionality, it is first converted into various data types, such as images [1–6], voxels [1,7–11], point clouds [12–14], and polygons [16–18,26]. Most of the studies utilize learning [1–3,7] with a CNN model for tasks such as searching, classification, segmentation, and generation, using data composed of 2D images or voxels.

In order to study the 3D environment in which we live, we collect data using various devices, such as cameras, 3D scanners, depth sensors, and motion sensors [16]. In order to express these extracted data in a virtual environment, they are transformed into the geometric data of a vertex and a face unit consisting of a vertex for expression.

In general, the geometry surrounded by the surface has the form of a vertex with position information, a face consisting of three vertices, and a polygon, a surface, and a geometry as a group of faces. According to the shape of the geometry, the number of vertices, the number of faces, and the number of faces sharing one vertex are configured differently, so it is difficult to structure it into generalized data. Since it is difficult to study geometric data due to their unstructured characteristics, practical polygon-based studies are rarely conducted. In this study, to overcome these limitations, we attempted to create end-to-end 3D geometric data through face-based data, which take the basic configuration of polygons; see Figure 1.

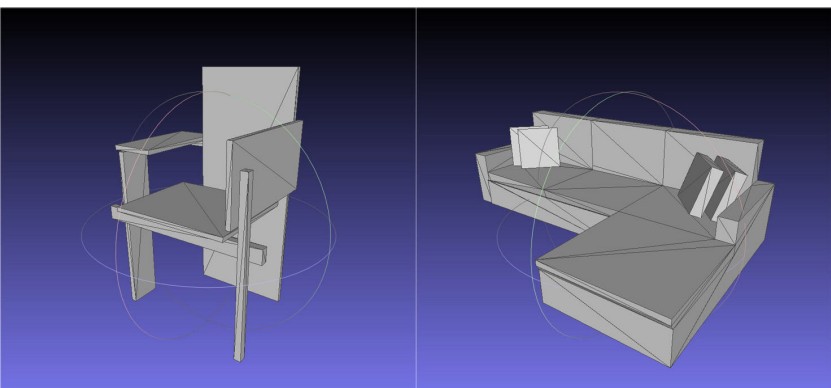

**Figure 1.** Face-based 3D geometry.

### 2.2. Graph Data

The relationship between vertices and faces in 3D data can be expressed as nodes and edges in graph data. Deep learning for graphs is being researched in various fields, such as social networks [19–22], chemistry [23,24], medicine [25], and computer vision [6,18,25]. The research closest [23] to our study succeeded in creating small graphs using variational autoencoders (VAE). However, 3D data that needs to represent a volume with surfaces has a much more complex structure than a graph data with an adjacency relationship between nodes and edges, and thus there are limitations in learning with the existing graph data structure. Accordingly, we propose a matrix structure suitable for 3D data in this study.

### 2.3. Generative Model

The most well-known generative models in unsupervised learning are approximate density-based variational autoencoders (VAE) [24,27–30] and implicit density-based generative adversarial nets (GAN) [31–33]. Recently, several papers, such as a study on generating an image through Multi-Adversarial Variational Autoencoder (MAVEN) combining GAN and VAE [34], and a study on generating graph data through Conditional VAE and Long Short Term Memory (LSTM) [35], produced results by combining VAE and other models. Among them, GAN was mainly used in studies [36] to train generative model using voxel data and Convolutional Neural Network (CNN) models, and VAE was mainly used in studies [20,23] to train using models such as graph data and Graph Convolutional Network

(GCN) [28]. In voxel data, one voxel has information similar to the surrounding voxel, so useful information can be extracted through a convolutional network, but one node of graph data has little data similarity with neighboring nodes. In addition, since graph data only has binary data on connection or not, we considered that it is possible to sufficiently learn through the VAE model.

In this study, the VAE generation model was used to learn and generate binary data easily and efficiently. All input data (Adjacency, Feature) was converted into binary data (0,1). We conducted research and experiments focusing on the most optimal 3D geometry data structurization and reconstruction for learning using VAE rather than the study of the VAE generation model itself. Through experiments, we have proven that the structurization of the optimal input data for the VAE generation model can achieve sufficient results.

## 3. Methods

For decades now, 3D geometric data has been used for product design, drawing, and simulation in many businesses. 3D geometric data are composed of a triangular face made of three points and a polygon with faces—see Figure 1—to realistically represent the shape. In order to express the shape more precisely and realistically, the face is composed of a smaller size and larger quantities and includes a normal vector, texture UV, and lighting map as additional information. However, for this expression, the number of vertices and faces varies depending on the object, and the location of vertices and the ways in which faces are organized are informal. For these reasons, it is not easy to study or learn 3D geometric data themselves, so research has been conducted by transforming these data into images [1–6], voxels [1,7–11], and point clouds [12–14]. We tried to confirm the possibility of learning by conducting research on 3D geometric data themselves, rather than on the existing deformation data. Through the experiment, we succeeded in learning face-based 3D geometry, and we confirmed its possibility.

As shown in Figure 2, we constructed a framework dedicated to geometry to learn polygonal geometric data. The first structurization is a step to structure unstructured geometric data into a structured data matrix and then train the structured matrix as input data through VAE, a generation model, and finally reconstruct the learned output data into geometry. Since it is not easy to grasp the reconstruction performance in the geometrical shape, in order to understand the learning performance, the initial geometry before structurization and the geometry that was reconstructed were respectively voxelized. By comparing the similarity of the corresponding voxel type, the performances for learning and generation were identified. Each step is described in more detail below.

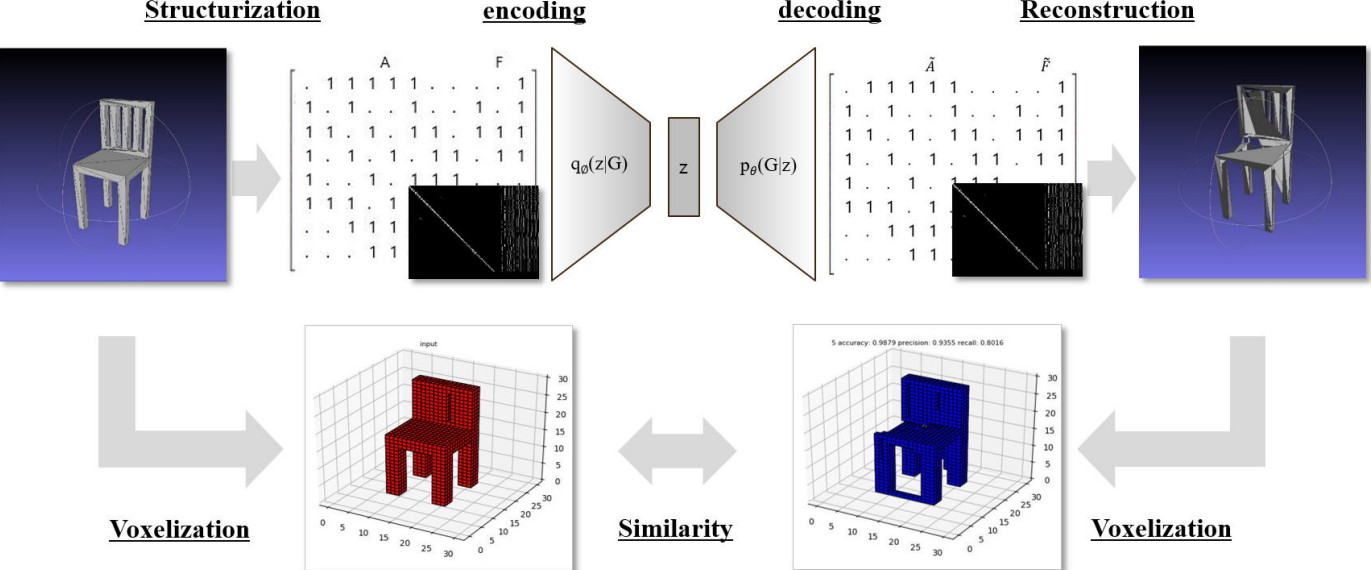

**Figure 2.** Framework of face-based variational autoencoders.

### 3.1. Adjacency Matrix Structurization

The structurization step for learning unstructured geometric data was the most important, and it was the most difficult step to find a formalization method for. Although the Adjacency||Feature (|| : Concatenating Matrices) (Equations (1)–(4)) of the existing graph data [19–21,23,25] is the most similar structure, it was not possible to express an optimized expression from its current form. Accordingly, three schemes for structurization optimized for 3D geometry were conceived. G in (Equations (1)–(4)) stands for geometry matrix.

$$G = (A, F) \tag{1}$$

**Vertex Adjacency ($A_V$).** In general, the geometry surrounded by the surface has the form of a vertex with position information, a face consisting of three vertices, and a polygon, a surface, and a geometry as a group of faces. Vertex and edge information connected to each other can be extracted from Geometry, and it has two to six connected vertices for forming a surface. In general, in the order of the vertex index, there is a connection relationship, and as shown in Figure 3, the vertex adjacency ($A_V$) matrix has diagonally sparse data. The size of the $A_V$ matrix (300) is greater than the number of vertices of the geometry with the most vertices (296), and it has a value of 1 if the horizontal vertex index and the vertical vertex index are connected, or 0 otherwise. The meaning of the horizontal and the vertical is the same, and it always has a symmetrical shape based on the diagonal. Unlike the existing graph data, as shown in Figure 3, the vertex adjacency ($A_V$) matrix has no value in the same horizontal and vertical index because the value of 1 represents the edge of two points. However, this matrix is expressed only with edge information; the surface information is missing and only the wireframe information is available.

$$G_{A_V+F_V} = (A_V, F_V) \tag{2}$$

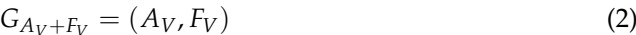

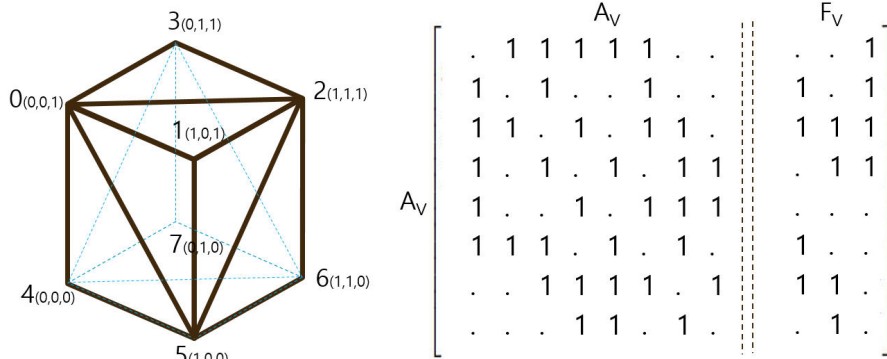

**Figure 3.** Vertex adjacency ($A_V$) and vertex feature ($F_V$ : position value of x/y/z) of a simple cube.

**Vertex*Face Adjacency ($A_{VF}$).** Since the geometry consists of a vertex and a face that shares the vertex, there is a limit to expressing the information using only the vertex adjacency. To solve this problem, a vertex*face adjacency ($A_{VF}$) matrix was conceived so that the vertex and face information can be simultaneously configured. The column of the adjacency matrix represents the vertex index, and the row represents the face index. Since one face always consists of three vertices, there are always three values in one face index column, as shown in Figure 4. Unlike the vertex adjacency, it can include face relationship information. However, the matrix efficiency differs depending on the ratio of vertex count and face count. This will be explained again in Section 4. Experiments.

$$G_{A_{VF}+F_V} = (A_{VF}, F_V) \tag{3}$$

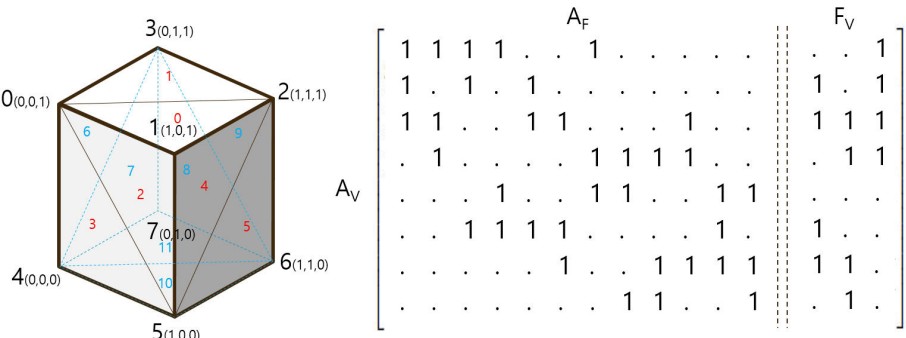

**Figure 4.** Vertex-face adjacency ($A_{VF}$) and vertex feature ($F_V$) of a simple cube.

**Face Adjacency ($A_F$).** The third method is to express the adjacency matrix consisting of faces that exist only in Geometry, not in a general vertex-centered graph structure. In general, a face is a triangle made of three vertices, and in a closed geometry, a triangle always has information connected to three other triangles. Like vertex adjacency, if the horizontal face index and the vertical face index share one edge, it has a value of 1 or 0 otherwise. Also like vertex adjacency, it has a symmetrical shape around the diagonal. However, since there are always three faces adjacent to one face, as shown in Figure 5, the face adjacency matrix ($A_F$) has a much more sparse characteristic than other adjacencies, and it is less efficient because it requires information about three vertices as a feature at the same time. For this reason, we proposed Face Adjacency ($A_F$) only in the methods section, we do not conduct experiments.

$$G_{A_F+F_F} = (A_F, F_F) \tag{4}$$

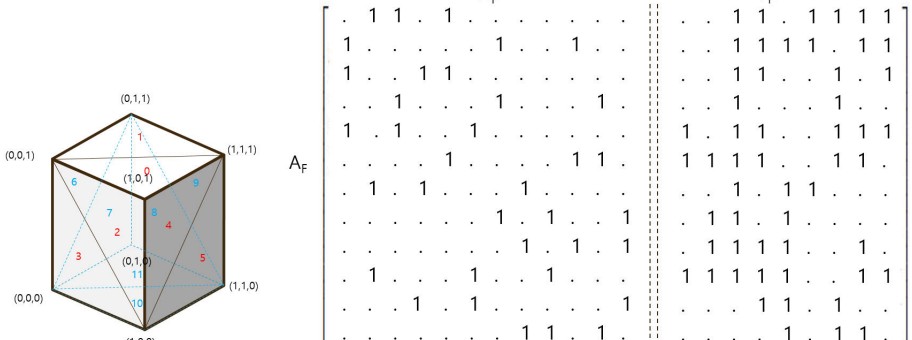

**Figure 5.** Face adjacency ($A_F$) and face feature ($F_F$: position value of 3 vertices x/y/z) of a simple cube.

### 3.2. Feature Matrix Structurization

**Vertex Feature ($F_V$).** Essentially, the most important information among vertex features in geometry is position (X,Y,Z). In order to generalize various sizes for each geometry type, the vertex position was generalized to a size of 0 to 1. All positions were expressed as real values to 4 decimal places (0.0001 to 0.9999) between 0 and 1. However, for accurate real value encoding, each digit number was expressed as one hot encoding. Accordingly, to store each value of X, Y, and Z, each of 40 binary values for a total of 120 vertex features were expressed; see Figure 6.

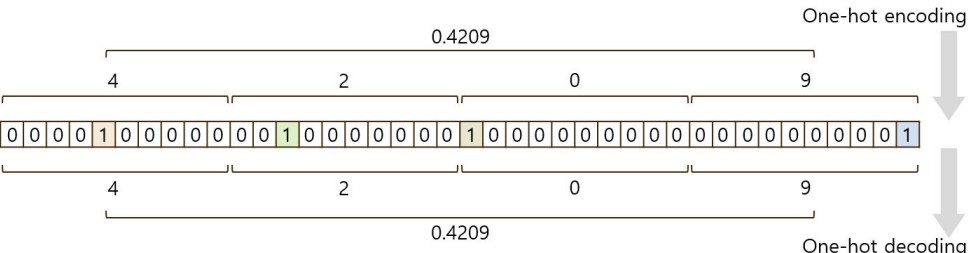

**Figure 6.** One hot encoding/decoding of vertex feature.

**Face Feature ($F_F$).** Since a triangle is specified by three vertices, it can include the position information of three points, the area of the face that can be extracted into three points, and the face normal vector as feature data. However, in this paper, for the purposes of simple comparison, feature information is conceived only with position information identical to the vertex feature. In addition, the generation model experiment for the face feature was excluded, as well as the face adjacency ($A_F$).

### 3.3. Face-Based Variation Autoencoders (FVAE)

We trained the generation model for graph-based geometric data using the existing variational autoencoder model [27,37]. The VAE model has the characteristic that the output value becomes blurry than the input value because of the reason that the result is generated through the small latent dimension of the mean and variance [38]. We converted all input data (adjacency, feature) into binary data (0, 1) so that we can achieve performance through the VAE model. In addition, we used an extended model (Face-Based VAE) with three hidden layers (with 1000 nodes) for encoding and decoding of the existing VAE model [27,37] to improve the learning performance Figure 7.

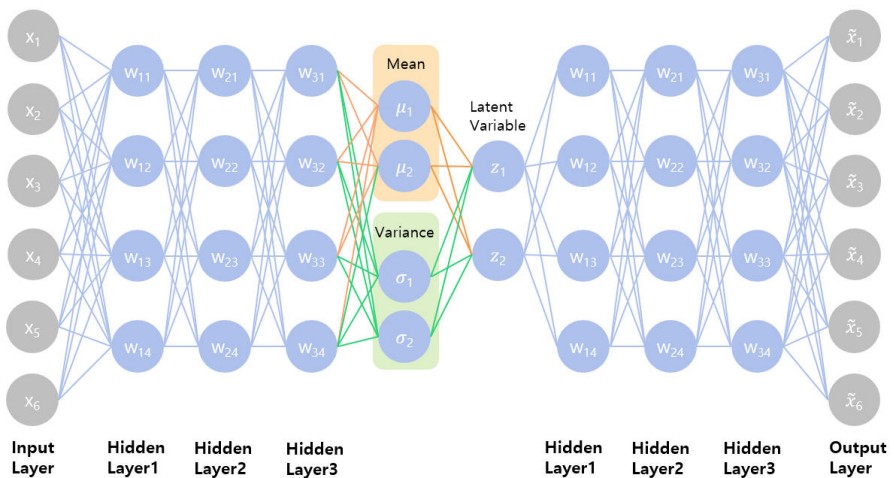

**Figure 7.** Architecture of face-based variational autoencoders.

Two types of input data were trained: a vertex-based adjacency matrix and a vertex*face-based adjacency matrix. Input value $G = (A, F)$ (1), A is the adjacency matrix of the vertex or face, and F is the position of one hot encoding data of the vertex. In Equation (5) of the total loss for reconstruction error ($E$) and regularization ($KL$), the input value x was replaced by G of Geometry. Except for the input value, the other equation is the same as in the previous study [27].

$$\mathcal{L}(\phi, \theta : G) = -E_{q_\theta(z|G)}[log(p_\theta(G|z))] + KL[q_\phi(z|G)||p(z)] \qquad (5)$$

As shown in Figure 8, the input data have information about vertex or vertex*face adjacency data in diagonal form and one hot encoding of the vertex position feature in

vertical line form. Since it has the geometric information in a fixed ($300 \times 370$: vertices data(300) × (faces data(250) + feature data(120))) matrix, there is a blank space depending on the sizes of the vertex and face. The issue of increasing voids in the matrix depending on the number of vertices and faces needs to be improved in future works.

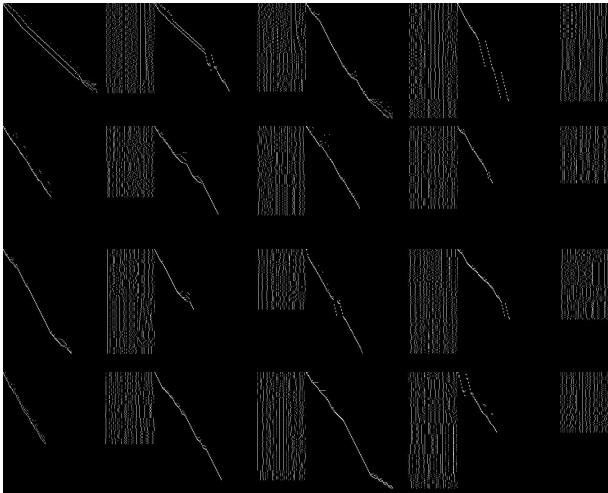

**Figure 8.** Input data of vertex-face adjacency ($A_{VF}$) and vertex feature ($F_V$) ($4 \times 4$).

*3.4. Reconstruction*

The step of reconstructing the geometric information from the structured output matrix was another challenge. In order to reconstruct the vertex of the geometry and the unstructured data of the face, the following solutions are suggested for each of the three matrices performed in the structurization step.

**Vertex Adjacency ($A_V$).** To create each piece of face information in an $A_V$ that contains only vertex and vertex edge information, it was necessary to find the total number of faces and the three points constituting each face. When we mapped the column and row values of 3 vertices of the cube blue triangle (Figure 9) in the vertex adjacency matrix ($A_V$), we found a rule for grouping the three points constituting the face (triangle) in the form of a right triangle. We found that the three points that make up this triangle are the three points that make up one face. Since the matrix has a symmetrical shape around the diagonal, the face information was extracted only at the top of the gray area. However, among the information was a case where some incorrect faces that did not exist as actual faces were created, such as the red triangle in Figure 9. However, it was impossible to distinguish incorrect information, such as a red face in the generated $A_V$ matrix, and since the face was an internal face in the overall shape, it was not a subject of great consideration from the perspective of the overall shape.

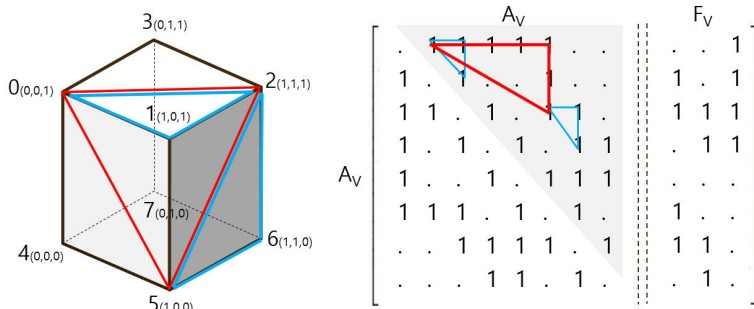

**Figure 9.** Reconstruction error of vertex adjacency ($A_V$) matrix.

**Vertex*Face Adjacency ($A_{VF}$).** As shown in Figure 4, in the $A_{VF}$ structure, vertex and face information can be clearly restored compared to $A_V$. The Geometric Object is

clearly reconstructed by extracting the three indexes of vertices from the vertical, which are organized according to the face index in the horizontal. However, it is distributed more sparsely than $A_V$ and is composed of an asymmetric shape rather than a horizontal and vertical square shape.

**Face Adjacency ($A_F$).** As can be seen in Figure 5, it is not easy to restore vertex index information because $A_F$ contains face-based information, not vertex-based information. We consider that the structure is more suitable for the task of supervised learning than the generative model. Accordingly, the generation model experiment for $A_F$ was omitted from this study.

*3.5. Voxelization*

It was not easy to evaluate the performance of the generated model using the face-based geometric data itself. There was too much information to be compared, such as the number of vertices, the X/Y/Z position to four decimal places of the vertex, the number of faces, and the composition of the vertex index within the face. Therefore, a similarity evaluation method that could be used as a standard was needed. To solve this problem, we evaluated the similarity by voxelizing the initial geometry and the generated geometry. Figure 10. This made it possible to compare objective and quantitative performance. If the face of each geometry passes through the corresponding voxel in the entire $30 \times 30 \times 30$ voxel space, or if there is a vertex in the voxel area, a value of 1 is entered, and if not, a value of 0 is entered. In order to measure consistent similarity for 3D geometries of various sizes, the bounding box of geometry was adjusted and transformed to a size that is full in voxel space. However, since the size ratio between x/y/z of geometry was maintained, a considerable proportion (over 80%) of voxels existed as empty spaces.

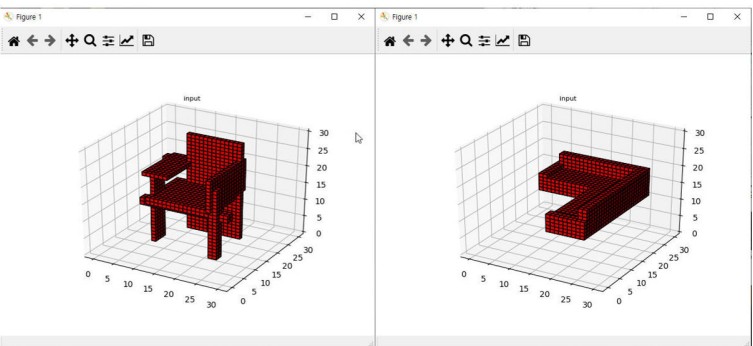

**Figure 10.** Voxelization of 3D geometry ($30 \times 30 \times 30$).

*3.6. Similarity*

In order to measure the similarity in this study, performance through the values of $Accuracy = (TN + TP)/(TN + TP + FN + FP)$, $Precision = TP/(TP + FP)$, and $Recall = TP/(TP + FN)$ was compared.

- True Positive (TP): Exists in the input voxel and exists in the output voxel;
- True Negative (TN): When it does not exist in the input voxel and does not exist in the output voxel;
- False Positive (FP): If it does not exist in the input voxel, but exists in the output voxel;
- False Negative(FN) : Exists in the input voxel and not in the output voxel.

However, since a large proportion of voxels in the voxel space ($30 \times 30 \times 30$) are empty spaces, the accuracy of using True Negative (TN) does not have a great meaning.

**4. Experiments**

The experiment was performed on five types of input data: $F_V$, $A_V$, $A_{VF}$, $A_V||F_V$, and $A_{VF}||F_V$. $F_V$ only learned about vertex features excluding adjacency, and $A_V$ and $A_{VF}$ only learned about adjacency excluding features. $A_V||F_V$ concatenates $A_V$ and $F_V$ to learn

about the entire matrix, and $A_{VF}||F_V$ concatenates $A_{VF}$ and $F_V$ to learn about the entire matrix. The source code was improved to optimize 3D geometry learning based on [37]. Tensorflow and Python packages (numpy, scipy, PIL, matplotlib) were used as libraries. The experiment environment was tested on a GPU (NVIDIA GeForce GTX 1070)-based i7-875H 2.20 GHz notebook, and the average value was measured by repeating each experiment 10 times (takes about 20 min at a time).

### 4.1. 3D Geometry Data

For the study of the 3D geometry generation model, Princeton Modelnet10 [2] was used as the data. Princeton Modelnet10 is divided into 10 categories for interior furniture and is a text file with only pure vertex position and face index information. In this study, for considering the performance of Multilayer Perceptron (MLP) learning and the system limitations, 64 geometry instances with less than 300 vertices were used as training data among about 900 Modelnet10 data, as shown in Figure 11, and 16 instances were used as test data. Even if only 300 vertices are converted to $A_V$ Matrix, 90,000 input nodes will be obtained. The number of input nodes on MLP used in the experiment was limited to 126,000 (300 × 420) of $G_{A_V+F_V}$.

### 4.2. Model

As shown in Figure 7, the structure of the model is in the form of MLP-based variational autoencoders. According to the $F_V$, $A_V$, $A_{VF}$, $A_V||F_V$, and $A_{VF}||F_V$ 5 experiments, each has 300 × 120(vertices × feature) , 300 × 300(vertices × vertices) , 300 × 250(vertices × faces) , 300 × 420(vertices × (vertices + feature)) and 300 × 370(vertices × (faces + feature)) input nodes. Encoding and decoding are composed of three hidden layers with 1000 nodes each.

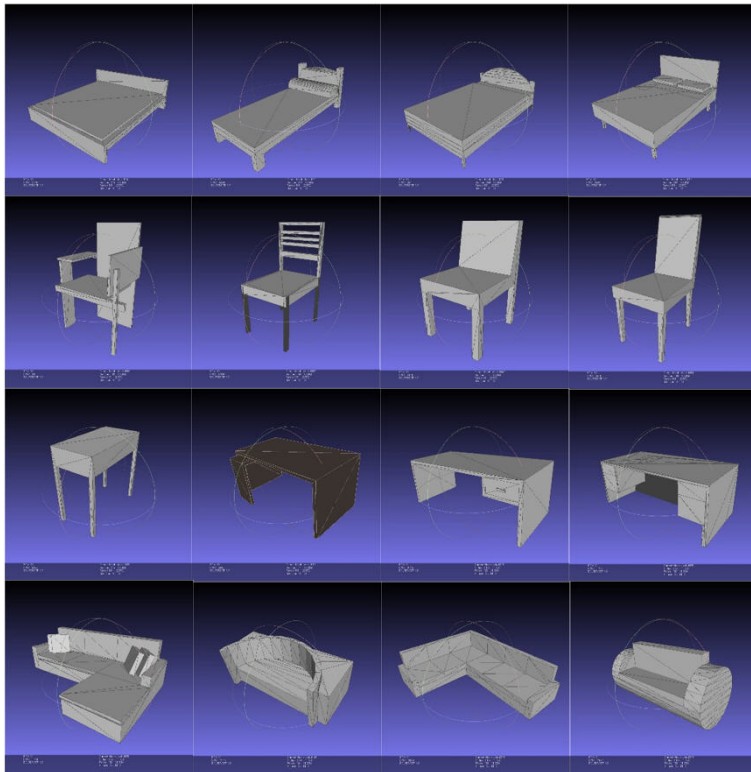

**Figure 11.** Princeton Modelnet10 data (under 300 vertices).

### 4.3. Generation

In the comparison of total training losses shown in Figure 12, it can be seen that $A_V$ and $A_{VF}$ learning-only adjacency learn much better than the $F_V$ learning-only feature. Here,

it was confirmed that the sparse adjacency matrix is more effective in learning than the one hot encoded feature information. Of $A_V||F_V$ and $A_{VF}||F_V$, it was confirmed that $A_{VF}||F_V$ was more effective in learning, and $A_{VF}$ with vertex*face information learned better than vertex-based adjacency.

Finally, the results learned while changing the latent variable $z$ value for the five experiments, $F_V$, $A_V$, $A_{VF}$, $A_V||F_V$, and $A_{VF}||F_V$, can be seen in Table 1. Basically, the accuracy was considered high in all experiments because the ratio occupied by true negative (TN) in the voxel space is high. For this reason, the performance was evaluated by the values of precision and recall rather than accuracy. The comparisons of $F_V/A_V/A_{VF}$, $A_V$ and $A_{VF}$ show much higher performance than $F_V$ in the precision (72.32, 63.48) and recall (69.86, 59.78) of test data. This confirms that it is much more difficult to learn position feature information than vertex and face adjacency information. It can be determined that the adjacent matrix is relatively well trained. Here, it can be seen that $A_V$ has a slightly higher performance on test data than does $A_{VF}$, which is considered due to somewhat overfitting during $A_{VF}$ learning because the number of input data is not large during training. In $A_V||F_V$ (z = 10) and $A_{VF}||F_V$ (z = 10), Precision (88.26, 99.99) and Recall (96.91, 100.00) of Training Data were high, but the reason for relatively low performance in Test Data is considered to be overfitting.

Looking at the performance results of $A_V||F_V$ and $A_{VF}||F_V$, it seems that it is difficult to draw a strong conclusion from the performance of test data yet, but it was confirmed that $A_{VF}||F_V$ is better trained than $A_V||F_V$ in training data.

Figure 13 shows the input geometry of test data and the generated output geometry for $A_V$. The dark face of the output geometry is caused by the reverse application of the normal vector direction of the face, but this research excluded the learning of the normal vector. Thus, it was confirmed that a fairly similar polygon geometry was created. Even if vertex position information is used as input data, that adjacency matrix data is generated with 72% Precision and 69% Recall performance is a great achievement.

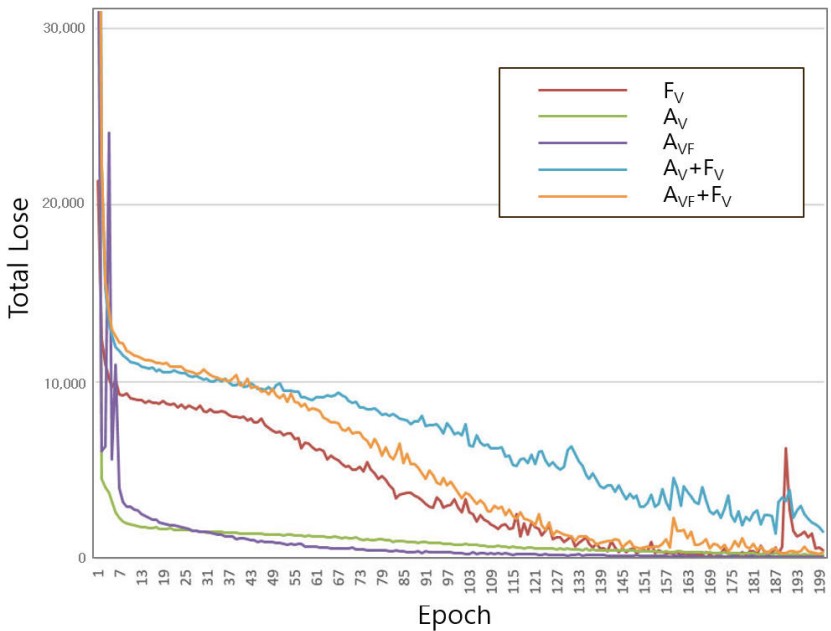

**Figure 12.** Comparison of total training losses.

**Table 1.** Performance on 3D geometry generation.

| G Type | Training Data(64) | | | Test Data(16) | | |
|---|---|---|---|---|---|---|
| | Accuracy | Precision | Recall | Accuracy | Precision | Recall |
| $F_V$ (z = 5) | 99.99 | 98.95 | 100.00 | 78.25 | 12.41 | 33.84 |
| $A_V$ (z = 5) | 99.72 | 97.82 | 98.64 | 95.84 | **72.32** | **69.86** |
| $A_{VF}$ (z = 5) | 100.00 | 100.00 | 100.00 | 94.08 | **63.48** | **59.78** |
| $A_V||F_V$ (z = 2) | 93.20 | 7.44 | 8.24 | 92.54 | 0.00 | 0.00 |
| $A_V||F_V$ (z = 5) | 88.92 | 49.01 | 71.27 | 81.80 | 10.91 | 8.63 |
| $A_V||F_V$ (z = 10) | 98.61 | **88.26** | **96.91** | 82.06 | 8.63 | 16.28 |
| $A_{VF}||F_V$ (z = 2) | 93.63 | 6.25 | 6.25 | 92.54 | 0.00 | 0.00 |
| $A_{VF}||F_V$ (z = 5) | 93.72 | 61.94 | 80.14 | 80.25 | 11.53 | 27.78 |
| $A_{VF}||F_V$ (z = 10) | 99.99 | **99.99** | **100.00** | 80.57 | 10.28 | 25.86 |

### 4.4. Geometry Data Analysis

We analyzed the data of 900 test data sets from Princeton Modelnet10 and found that the face/vertex ratio value was concentrated to a value of less than 1, as shown in Figure 14. This shows that most of the geometries consist of fewer faces than the number of vertices. This analysis demonstrates that in terms of data efficiency, $A_{VF}$ can perform structurization through a smaller matrix size than $A_V$ and learning proceeds better.

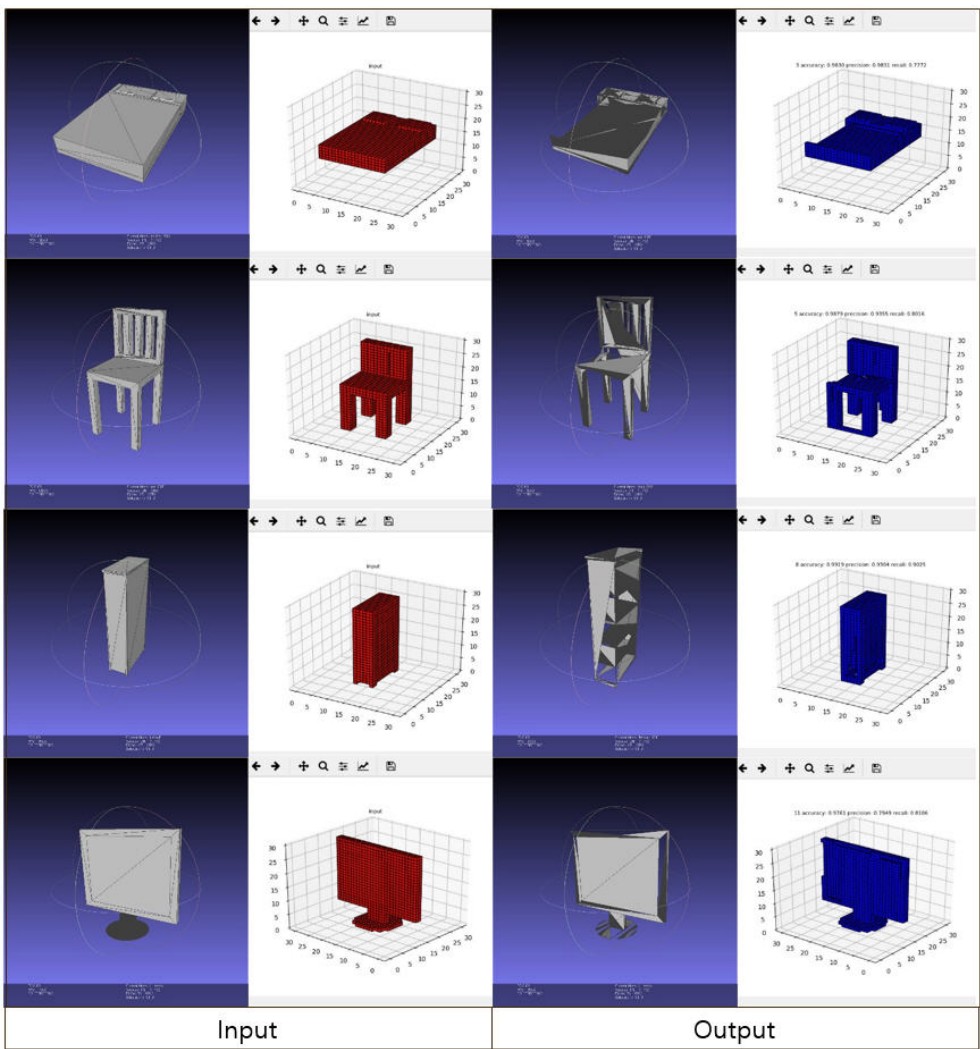

**Figure 13.** Representation of $A_V$ test data generation.

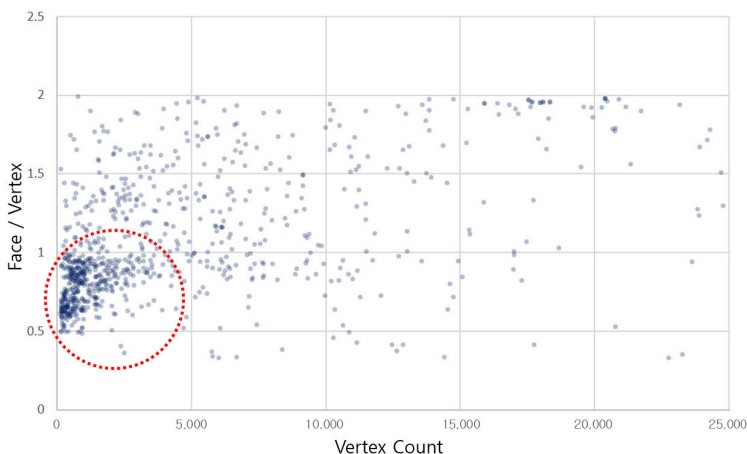

**Figure 14.** Face/vertex ratio according to vertex count.

## 5. Conclusions

In this research, we presented a face-based variational autoencoders (FVAE) model that generates 3D geometric data. Unlike the existing voxel-based research, our model conducts end-to-end learning using face information, which is the basic configuration of polygons actually used in industrial sites. The existing node- and edge-based adjacency matrix was improved and optimized for geometric learning using a face- and edge-based adjacency matrix according to a 3D geometric structure, and its performance was verified. The performance of the test data was not as high as expected, creating polygon data directly from end to end with 72% precision and 69% recall is a great achievement.

The contributions of this study are as follows:

- We presented a face-based 3D geometry generation model that directly generates polygon data from end-to-end without data conversion.
- We achieved the result of generating adjacency matrix information with 72% precision and 69% recall through end-to-end learning of Face-Based 3D Geometry.
- We presented various structurization methods for 3D unstructured geometry and proved the method to effectively perform reconstruction of the learned structured data.

**Future Works.** In this study, in consideration of the performance of MLP learning, learning was conducted only for geometries with 300 or less vertices. Of the 900 test data sets of the entire Princeton Modelnet10, 64 geometries (training data) were too small to form sufficient manifold space. In order to improve this situation, further research is needed on the processing plan for large-capacity vertex geometry. To accomplish this, additional research should utilize the concept of octree, a traditional CAD query method. Through sufficient training data, we can expect improvement of learning effect and generation performance.

**Author Contributions:** Conceptualization, S.P. and H.K.; writing—original draft preparation, S.P. and H.K.; writing—review and editing, S.P. and H.K. All authors have read and agreed to the published version of the manuscript.

**Funding:** This research was supported by the MSIT(Ministry of Science and ICT), Korea, under the ITRC(Information Technology Research Center) support program (IITP-2018-0-01405) supervised by the IITP (Institute for Information & Communications Technology Planning & Evaluation).

**Conflicts of Interest:** The authors declare no conflict of interest.

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
