# Peer review of "FaceVAE: Generation of a 3D Geometric Object Using Variational Autoencoders"

_electronics, doi:10.3390/electronics10222792_

Round 1
Reviewer 1 Report
Interesting work, I would like to thank the authors very much. Please consider my feedback below for the next version.
(1)
In the introduction, there is a dire need to cite references in order to support multiple statements mentioned. Please add references to support statements such as:
“the research is being conducted mainly on 2D images, voxels, and point clouds, which are relatively easy to learn”
“there is a limit to directly applying the research results in the environment of the polygon method used in actual industrial sites”
“research on direct polygon-based data is required”
“In deep learning about graphs, much research has been done in the fields of social networks, chemistry, medicine, and computer vision, among others, and many achievements have been made”
“Deep learning’s effectiveness has been proven in link prediction and label discrimination”
(2)
The literature review needs to be extended with referring to more recent contributions. Further, it is important to mention recent studies that applied VAEs for generative modeling, especially using image/graph data. For example:
https://doi.org/10.1007/978-981-15-6759-9_11
https://doi.org/10.3390/jimaging7050083
https://doi.org/10.1109/ICDCS51616.2021.00119
(3)
I encourage the authors to mention the key contribution(s) more confidently in the introduction and the critique of literature review. The contributions should be positioned clearly in line with the literature. Please make it clear if the study presents a novel application of VAE and/or other aspects of the methodology.
(4)
While GANs are becoming more and more popular, I find it important for the study to elaborate further on why the authors preferred the VAE approach.
(5)
If possible, please mention the running time required to train the VAE model. It is also worth mentioning more specific info on the GPU used.
(6)
Please mention all the libraries used (e.g. TensorFlow), and please cite their references (GitHub repo or publication).
Reviewer 2 Report
The paper describes a variational autoencoder (VAE) that uses face-based (triangle-based) geometry data to generate 3D geometry. The approach seems to be promising but it should be better described, with precise terminology and notation, and on the larger data set. More specific comments and suggestions are provided below.
Line 39: Instead of "extract" consider using "collect".
Line 43: Elaborate what do you mean by "unstructured characteristics".
Line 48: "The relationship between the vertex and face of 3D data is composed of a graph." Not sure what is the meaning of this sentence. Do you mean that a graph captures the relationship between the vertices and the faces in the geometry?
Line 52: "3D data has a much more complex structure than a graph" Elaborate. Why bipartite, multipartite or hyper graphs are not sufficient?
Line 61: Describe acronyms when first used. I assume GCN stands for "Graph Convolutional Network"
Lines 61-63: "This is understood because a graph has little data relevance to adjacent nodes, and the binary data has a strong nature of connection." Not sure what the authors are trying to say, please rephrase/elaborate.
Line 66: Instead of "industrial sites" consider using "business".
Line 79: Figure 1 referenced after Figure 2.
Line 91: What is (1)? (Equation 1)? Also, explain notation "Adjacency || Feature".
Line 94: What is (1)? (Equation 1)? Please clarify what do you mean by "G has been replaced by Geometry in the existing graph."
Line 101: "AV matrix" shouldn't V be a subscript? Also if the size is more than the number of vertices, what exactly is the size? Be specific.
Figure 3: Be clear that the graph includes face diagonals. If so, the diagonal connecting vertices 0 and 2 should be a thick line since it is visible. That way it would be clear why vertices (row/columns) 0, 2, 4, and 6 have five "1" and the other have three "1". Also, clarify what is F_{V}. There are three columns. Shouldn't there be six columns, one for each cube's face (square)?
Lines 132-133: "vertical column" Columns are by definition vertical.
Line 133: "horizontal column" use "row" instead.
Figure 4: Why there are only 4 columns for F_{V}, a cube has six faces (squares), not four?
Line 125: "there are always three faces adjacent to one face" that works only if faces are triangles and the geometry is closed, i.e., there are no boundary vertices. Please elaborate. If (line 121) you consider that a face is a triangle, then use triangle instead of face in the subsequent text. Referring to Figure 4, if face is always a triangle, then there should be 12 columns for F_{V} and each column should have three "1". Please clarify.
Line 128-129: "we only proposed structurization and excluded it from the experiment." Please clarify what that means.
Figure 5: Explain what is F_{F}. Why are there are nine columns? For example, the first column has "1" for faces (triangles) for faces 4, 5, and 9. Please clarify.
Line 139: "Since a face basically consists of three points" should be "Since a triangle is specified by three vertices".
Equations 2, 3 and 4: they are not referenced in the text. Why then providing equation numbers?
Equation 5: provided without proper context, discussion or explanation.
Figure 6 :needs better description. E.g., why 300*370 dimensions? Relate it to the values in line 231.
Line 167: "right-angled triangle" do you need to know that an angle is a right-angle to do reconstruction? That seems very limiting.
Line 176: "AVF structure" shouldn't VF be a subscript?
Line 210: "However, since the proportion of true negative (TN) occupies a large part of the voxel space (30*30*30) in this study, the accuracy could not demonstrate great meaning." but in lines 199-120 it is stated "the geometry was adjusted and transformed to a size that is full in voxel space." Please clarify.
Line 224: Describe acronyms when first used. I assume MLP stands for "Multilayer Perceptron"?
Line 225: The upper limit is 300 vertices. Why? Was that influenced by system performances?
Line 245: "FV" shouldn't V be a subscript?
Table 1: Explain why for some values bold style is used.
Line 260: "recall by more than 70% is a great achievement." Table 1 for test data show recall less than 70%. Please clarify.
Line 278: This is not a framework in a full sense. Different forms of the input geometry data for a VAE were used and the results compared.
Lines 284-285: "less than 300 geometries were too small to form sufficient manifold space." Please clarify.
Round 2
Reviewer 1 Report
Thanks very much for your response to the feedback.